# Development and Validation of the Interprofessional Collaboration Practice Competency Scale (IPCPCS) for Clinical Nurses

**DOI:** 10.3390/healthcare12070806

**Published:** 2024-04-08

**Authors:** Yen-Fang Chou, Suh-Ing Hsieh, Yi-Ping Tseng, Shu-Ling Yeh, Ming-Chu Chiang, Chia-Chi Hsiao, Chiu-Tzu Lin, Shui-Tao Hu, Sue-Hsien Chen, Mei-Nan Liao

**Affiliations:** 1Department of Nursing, Chiayi Chang Gung Memorial Hospital, Chiayi County 61363, Taiwan; yenfc@cgmh.org.tw (Y.-F.C.); chiachi@cgmh.org.tw (C.-C.H.); 2School of Nursing, College of Nursing, Taipei Medical University, Taipei City 11031, Taiwan; 3Department of Gerontology and Health Care Management, Chang Gung University of Science and Technology, Taoyuan City 33303, Taiwan; 4Department of Nursing, Chang Gung University of Science and Technology, Taoyuan City 33303, Taiwan; 5Department of Nursing, Taoyuan Chang Gung Memorial Hospital, Taoyuan City 33378, Taiwan; sarah0413@cgmh.org.tw; 6School of Nursing, College of Medicine, National Taiwan University, Taipei City 10617, Taiwan; 7Department of Nursing, Keelung Chang Gung Memorial Hospital, Keelung City 20401, Taiwan; q22122@cgmh.org.tw (S.-L.Y.); sheri0930@cgmh.org.tw (S.-T.H.); 8Department of Nursing, Kaohsiung Chang Gung Memorial Hospital, Kaohsiung City 83301, Taiwan; e2988386@cgmh.org.tw; 9Department of Nursing, Linkou Chang Gung Memorial Hospital, Taoyuan City 33305, Taiwan; q22208@cgmh.org.tw; 10Administration Center, Chang Gung Medical Foundation, Taoyuan City 33305, Taiwan; a66998@cgmh.org.tw (S.-H.C.); jy46912@cgmh.org.tw (M.-N.L.); 11Department of Nursing, Chang Gung University, Taoyuan City 33375, Taiwan

**Keywords:** interprofessional collaborative practice, competency, scale, psychometric study, clinical nurse, exploratory factor analysis, principal-axis factoring

## Abstract

Interprofessional collaborative practice is a core competency and is the key to strengthening health practice systems in order to deliver safe and high-quality nursing practice. However, there is no Interprofessional Collaboration Practice Competency Scale (IPCPCS) for clinical nurses in Taiwan. Therefore, the purposes of this study were to develop an IPCPCS and to verify its reliability and validity. This was a psychometric study with a cross-sectional survey using convenience sampling to recruit nurses from the seven hospitals of a medical foundation. A self-designed structured IPCPCS was rolled out via a Google survey. The data were analyzed using descriptive statistics, principal-axis factoring (PAF) with Promax rotation, Pearson correlation, reliability analysis, and one-way ANOVA. PAF analysis found that three factors could explain 77.76% of cumulative variance. These were collaborative leadership and interprofessional conflict resolution, interprofessional communication and team functioning, and role clarification and client-centered care. The internal consistency of the three factors (Cronbach’s α) was between 0.970 to 0.978, and the Pearson correlation coefficients were between 0.814 to 0.883. Significant differences were presented in the IPCPCS score by age, education level, total years of work experience, position on the nursing clinical ladder, and participation in interprofessional education. In conclusion, the three factors used in the IPCPCS have good reliability and construct validity. This scale can be used as an evaluation tool of in-service interprofessional education courses for clinical nurses.

## 1. Introduction

The ultimate goal of modern healthcare is to deliver patient-centered, holistic care of the highest quality. Interprofessional collaborative practice (IPCP) has emerged as the premier strategy for achieving this goal. The Institute of Medicine (IOM), in its 2003 report, underscored the importance of IPCP in enhancing patient safety and healthcare quality [1]. This focus was amplified by the IOM’s 1999 report, “To Err is Human”, which identified medical errors as a major global healthcare challenge, thereby stressing the need for safer healthcare practices [2]. A pivotal change required in healthcare institutions is the promotion of interprofessional collaboration. Training healthcare professionals in teamwork is essential in efforts to foster interprofessional collaborative care, which in turn improves patient safety [2].

The IOM has delineated five core competencies necessary for healthcare professionals to deliver standard medical care: patient-centered care, evidence-based practice, information technology, quality improvement, and interprofessional teamwork [1]. IPCP fosters communication across different professional domains, facilitating mutual learning and the development of innovative problem-solving strategies. This approach promotes a patient-centered care ethos, minimizes resource wastage, and enhances the quality of medical care [3].

Furthermore, in 2010, the World Health Organization (WHO) defined IPCP as a collaborative effort involving healthcare workers from diverse backgrounds, along with patients, families, caregivers, and communities, with an aim to provide comprehensive, high-quality care [4]. This innovative model necessitates a clear understanding of each participant’s role, encouraging learning from and contribution to the team, as well as the sharing of responsibilities, and is considered the core and soul of healthcare [5,6].

IPCP plays a crucial role in improving the accessibility of health interventions, strengthening interdepartmental coordination, and enhancing the job satisfaction of healthcare professionals [4,5]. For patients, IPCP has numerous positive health-related outcomes. According to a scoping review by Lutfiyya et al. (2019) [7], these benefits include well-managed chronic diseases, reduced surgery costs, improved communication between care providers, and increased patient satisfaction, among others [7]. For nursing staff, effective teamwork and communication can increase job satisfaction [8].

The identification of competencies, as highlighted by Ten Cate O (2006), is a crucial initial step in preparing future healthcare providers to deliver high-quality care [9]. In response, the World Health Organization, along with the Interprofessional Education Collaborative (IPEC) and the Canadian Interprofessional Health Collaborative (CIHC), proposed core IPCP competencies for adoption by medical education-related research institutions [1,5,10,11]. IPEC’s four IPCP core competencies are values/ethics for interprofessional practice, roles/responsibilities, interprofessional communication, and teams and teamwork [1,10]. The six dimensions proposed by CIHC are role clarification (RC), patient/client/family/community-centered care (CC), interprofessional communication (IC), team functioning (TF), collaborative leadership (CL), and interprofessional conflict resolution (ICR) [5,11].

Working based on the concepts of IPEC or CIHC, many international scholars have continued to develop IPCP assessment tools. The assessment targets different clinical practitioners or single professions, diverse clinical practitioners and healthcare professional students, and students in different healthcare professions. The scale is mainly used for self-assessment but rarely for observer use in the assessment of clinical practitioners. Even the IPCP assessments among different clinical practitioners or patients within an organization are also scarce (Appendix A) [12,13,14,15,16,17,18,19,20,21,22,23,24,25,26,27,28,29,30,31,32,33,34,35,36,37,38,39,40,41,42,43,44,45,46,47]. Furthermore, there are several outcome measurement instruments for evaluating competency in interprofessional collaboration practice [48,49]. These IPCP assessment tools, depending on their measurement purposes, are used in different healthcare contexts and for different subjects. Although each tool has advantages and disadvantages, earlier tools had methodological issues related to psychometric tests. For instance, Peltonen et al. (20) pointed out in their scoping review that the psychometric parameters of 29 tools were non-systematic, focusing primarily on construct validity and internal consistency. They suggested that further extensive testing and confirmatory studies should be performed to strengthen the evidence of the reliability and validity of these tools [48]. Moreover, Glover et al. (2022) identified similar issues in their systematic review of outcome measurement instruments. Only ATICS-II was rated as having sufficient measurement characteristics, with low- to moderate-quality evidence, and further validation of each outcome measurement instrument, including relevance, comprehensibility, and comprehensiveness, is needed [49]. Considering the differences in healthcare contexts and cultures across countries, Edelbring et al. (2018) argue that caution should be exercised in the conceptualization and wording of scales during localization and translation [43].

Cultivating competence in “interprofessional collaborative practice” and examining the implementation of IPCP among healthcare personnel are the best educational and summative assessment methods for interprofessional education (IPE) [50], leading to the best outcomes in terms of patient-centered care [1]. Since 2007, Taiwan’s Ministry of Health and Welfare has promoted a subsidy program for teaching hospitals to train novice medical personnel in IOM core competencies. Undergoing continuous medical-quality education in teaching hospitals, particularly in the “Two-Year Nurse Post-Graduate Year (NPGY) Training Program” [51], mandates participation in interprofessional team collaboration care-related activities (IPE) during the second (4–12 months) and third stages (13–24 months) of the program. This policy fosters the effective implementation of IPCP, laying the groundwork for Taiwanese medical care institutions to aim for patient-centered holistic medical care. Although the importance of IPCP in medical care has been noted, there is still a lack of valid and reliable tools for assessing IPCP competencies among clinical nurses on different clinical nursing ladders in Taiwan. The dimensions of leadership and conflict management are missed by many IPCP scales [12,13,14,15,16,17,18,19,20,21,22,23,24,25,26,27,28,29,30,31,32,33,34,35,36,37,38,39,40,41,42,43,44,45,46,47]. Since the importance and necessity of IPCP competencies among nursing staff are acknowledged, clinical nurses, as members of interprofessional teams, should possess IPCP capabilities [42], promote patient safety and care quality [7], and achieve the goal of holistic care as a crucial aspect of nursing practice. Considering these challenges, our study aims to develop and validate a theoretically based self-assessment tool for evaluating the “Interprofessional Collaborative Practice Competency” among clinical nurses in Taiwan. This initiative addresses the critical need for a valid and reliable IPCP competency assessment tool in the Taiwanese healthcare context, supporting the advancement of patient safety and care quality through enhanced nursing practice.

## 2. Materials and Methods

### 2.1. Study Design

This was a psychometric study with a cross-sectional survey using a self-designed structured scale.

### 2.2. Participants and Setting

The study was based on a convenience sample of clinical nurses from seven hospital districts of a medical foundation in Taiwan. The inclusion criteria were registered nurses who were willing to fill out the questionnaire and agreed to participate in the study program. The exclusion criteria were medical personnel in non-nursing occupations. The sample size was calculated using the Creative Research System (2012) sample-size estimator with a confidence interval set at 95%, a confidence level set at 4, and a population of 566 based on approximately 10,000 registered professional nurses [52].

### 2.3. Instruments

The tools used in this study consisted of two parts. The first part was sociodemographic and professional characteristics, including gender, age, education level, hospital district, years of work experience, place on the nursing clinical ladder, qualification as a clinical instructor, and training experience in interprofessional education (IPE). The second part was the score achieved on the Interprofessional Collaborative Practice Competence Scale (IPCPCS) based on the six competency structures of the CIHC [5,11], as shown in Appendix A. After receiving approval from the CIHC team, two experts with nursing backgrounds and doctoral degrees, who had lived in English-speaking countries for more than 5 years, conducted English-to-Chinese forward translation and Chinese-to-English back translation of the questionnaires [53,54], considering the cultural and language differences between Chinese and English questions. The original IPCPCS has 41 questions with 6 dimensions, including role clarification, patient/client/family/community-centered practice, interprofessional communication, team functioning, collaborative leadership, and interprofessional conflict resolution. A 7-point Likert scale was used, with responses on a scale with 1 indicating “No ability”; 2 indicating “Inadequate ability”; 3 designating “Somewhat inadequate ability”; 4 indicating “Ability satisfied or meeting minimal requirements”; 5 indicating “Fair ability”; 6 showing “Good ability”; and 7 sjowing “Excellent ability”. Possible total scores ranged from 42 to 210, with higher scores indicating greater holistic practice competence in terms of interprofessional collaboration.

After the questionnaire was drafted, a total of five experts in related fields in Taiwan (1 physician, 1 associate professor, 1 director of the Nursing Department, and 2 experts specializing in nursing and health practice education) were invited to perform expert content validity evaluation. The experts rated the relevance of the questions to the study’s purpose by using a four-point Likert scale (4 = appropriate; 3 = requiring minor modification; 2 = requiring major modification; and 1 = inappropriate) to calculate the content validity index (CVI) and quantify expert validity [53]. The questionnaire was modified according to the expert’s recommendations and the modified research tools had 41 questions.

### 2.4. Procedure

Figure 1 presents the flow of this study. Based on the literature [53,54], the English-to-Chinese and Chinese-to-English translations took cultural differences into consideration. A 41-question survey was constructed along six dimensions. Five experts were commissioned to conduct an expert content validity evaluation, and the content of the questions was modified according to their comments. After obtaining approval from the IRB, a pilot test was conducted in a hospital district. As there were no changes in the content of the questionnaires, the project investigator (PI) of the study sent a study recruitment poster and a QR code link to the responsible supervisors of seven hospital districts. After obtaining approval from supervisors, the poster and link were forwarded to the head nurses, and then the head nurses distributed the study’s instructions and the survey QR code URL to the clinical nurses. The Google online link-based questionnaire was distributed from 5 February 2022 to 30 November 2022 to conduct a formal test. Then, the reliability and validity of this tool were analyzed.

### 2.5. Ethical Considerations

This study was conducted in compliance with the basic human rights of the participants, safety standards, and research ethics and was approved by the Institutional Review Board of the Chang Gung Medical Foundation (202102065A3). Consent was obtained from all participants. Personal information was handled with anonymous numbering and kept strictly confidential to protect the privacy of the subjects.

### 2.6. Data Processing and Statistical Analysis 

The results of the questionnaire were automatically saved in Google Forms and downloaded into the 2016 version of our Excel database after the study’s deadline date to examine the accuracy of the data. The SPSS 22.0 Statistical Package (IBM, Armonk, NY, USA) was used for the assumption testing and data analysis. Assumption testing included normality, linearity, outliers, and multicollinearity. Data analysis used descriptive statistics to present the sociodemographic and professional characteristics of participants and the distribution of scales in terms of value, percentage, mean, median, and standard deviation. Reliability analysis was performed to assess the internal consistency of the IPCPCS. Principal-axis factoring analysis was conducted with Promax rotation to determine validity based on initial eigenvalues (≥1), factor loadings (>0.40), and scree plots [55,56,57]. One-way analysis of variance (ANOVA) was applied to check the mean differences between the total IPCPCS scores by examining the sociodemographic and professional characteristics of the participants, with a *p* < 0.05 two-tailed test result indicating statistical significance. Pearson correlation was applied to analyze the relationship between the factors for the IPCPCS.

## 3. Results

In this study, 578 questionnaires were collected. The fill-in rate was 97.9%. Winsorizing was used to remove 5 extreme values, and the final statistical analysis was conducted with 548 valid questionnaires.

### 3.1. Sociodemographic and Professional Characteristics of the Participants

Most of the participants were female (96.7%). The most common age group was 26 to 30 years (22.6%), with a mean age of 34.48 (SD = 7.87). Moreover, 84.3% of the participants had a bachelor’s degree in nursing. The most common work experience group was 5.1 to 10 years (29.9%), with a mean of 11.08 (SD = 7.80). Most of the participants (25.0%) were of rank N2. Of all the nurses, 74.3% had clinical teaching qualifications, and 51.1% had experience with IPE, as shown in Table 1.

### 3.2. Content Validity

The S-CVI of this research tool was 0.76 and its I-CVI was 0.94.

### 3.3. Construct Validity—Principal-Axis Factoring of the Exploratory Factor Analysis

We employed principal-axis factoring analysis for factor extraction of the scale, followed by a Promax rotation. After the sequential removal of one item due to multicollinearity (ICR5) and two items with factor loadings below 0.40 (TF1 and TF3), the final scale comprised 38 items. The overall Kaiser–Meyer–Olkin (KMO) measure of sampling adequacy was exceptionally high at 0.983, and Bartlett’s test of sphericity was significant (χ^2^_(703)_ = 29, 303.47, *p* < 0.001). The measure of sampling adequacy (MSA) of individual items ranged from 0.975 to 0.989. The scree plot illustrates three factors (Figure 2).

Table 2 shows that the eigenvalues of Factor 1 to Factor 3 after Promax rotation were 26.26, 2.16, and 1.13, respectively, explaining 69.11%, 5.69%, and 2.97% of the variance. The explainable cumulatively variance was 77.76%. As delineated in Appendix B (Table A1), the three factors of the scale were termed: Factor 1—collaborative leadership and interprofessional conflict resolution (CLICR, 13 items); Factor 2—interprofessional communication and team functioning (ICTF, 12 items); and Factor 3—role clarification and client-centered care (RCCC, 13 items).

### 3.4. Construct Validity—Contrasted Groups Methods

Assessed using ANOVA, age (*p* = 0.001), education level (*p* = 0.002), total years of work experience (*p* < 0.000), nursing clinical ladder (*p* < 0.000), and prior participation in an interprofessional education (*p* < 0.000) by clinical nurses produced significant differences in the IPCPCS score (Appendix B, Table A2).

### 3.5. Internal Consistency Reliability

Table A1 (Appendix B) shows that Cronbach’s α for the three factors and the overall scale was greater than 0.970, with Cronbach’s α of the overall scale being 0.988. Table 3 displays Pearson’s two-tailed correlation analysis, showing significant positive correlations among the three factors (*p* < 0.01).

## 4. Discussion

The primary aim of this study was to develop and validate the Interprofessional Collaborative Practice Competency Scale (IPCPCS). This study was grounded in the Canadian Interprofessional Health Collaborative (CIHC) framework, which outlines six major areas of competency regarding interprofessional collaboration. The process of scale development encompassed the confirmation of the core competency framework, the design of scale items, and the assessment of content validity, construct validity, and internal consistency. The results validated the scale as a reliable and effective tool for assessing nurses’ IPCPCS scores.

### 4.1. Sociodemographic and Professional Characteristics of the Participants

The participants in our study were all clinical nurses, with an average age of 34.48. Most of the participants had a bachelor’s degree, and the average years of work experience was 11.08. Approximately 63% of the nurses held level N3 or above, and the majority of participants were qualified as clinical nursing instructors (74.3%). However, only 51.1% had participated in IPE training, indicating the necessity of IPCP training regardless of work experience.

### 4.2. Content Validity

For content validity, the appropriateness of each item content and direction was assessed by experts, and the results achieved an ideal level (S-CVI = 0.76 & I-CVI = 0.94), indicating good individual item values and the overall validity of the scale’s content [54].

### 4.3. Construct Validity—Principal-Axis Factoring of the Exploratory Factor Analysis

Construct validity was examined using principal-axis factoring from exploratory factor analysis with Promax rotation. The overall Kaiser–Meyer–Olkin (KMO) measure of sampling adequacy, individual Measures of Sampling Adequacy (MSA), and Bartlett’s test of sphericity were significant, indicating that the data were highly suitable for use in factor analysis [57]. The factor loadings of the items in the scale ranged from 0.44 to 0.97. For exploratory factors, the minimum standard for factor loading should be greater than 0.3, and a score above 0.55 is considered good [56].

In the literature, the validation of IPCP competencies predominantly references the IPEC and CIHC competency frameworks. Our analysis indicates that the various dimensions of interprofessional collaboration competencies, initially encompassed by six dimensions, were reduced to three factors after factor analysis, which is supported by other scholars’ research findings [5,11,18,21,26,28,39,43,44,47]. Our study found that nurses’ self-assessment based the three dimensions of the IPCPCS were highest in ICTF, which is followed by RCCC, and lowest in CLICR. Unlike IPC [22], CICS29 [25,30], JASSIC [34], and IPC-Thailand [32], the factors of leadership and conflict management were not mentioned. Recent empirical literature found that clear role functions and leadership, which have significant implications for IPCP core competencies, are influential factors in effective team collaboration [59]. Nursing education should incorporate basic concepts of IPCP core competencies, such as leadership and management skills in clinical nursing education. Understanding roles and responsibilities in IPCP is crucial for enabling novice nurses to perform leadership roles within collaborative teams, advocate for patients, and promote patient safety and care quality [60,61].

### 4.4. Construct Validity—Contrasted Groups Methods

The study found significant differences in IPCPCS values among nurses based on age, educational level, total years of work experience, clinical nursing ladder, and IPE participation. The differences may be due to nurses being aged ≥ 46 years, holding senior positions or being higher up the nursing ladder, and having participated in the IPE, resulting in their scoring higher in the IPCPCS. Nurses with postgraduate degrees were caring and possessed critical thinking and reasoning aptitudes, professional nursing skills, communication and teamwork abilities, ethics, the attribute of responsibility, and lifelong learning capabilities. Therefore, they had higher IPCPCS scores. The literature confirms that learning experiences, age, participation in IPE training, and management experience positively affect the IPCP [62], and scholars point out that interprofessional education (IPE) programs are the first step in implementing the IPCP [63]. Recent intervention studies have shown that interprofessional simulation education can help novice nurses to improve core nursing competencies [64]. Moreover, scholars have noted that current nursing educators are still underprepared for teaching IPCP, which affects the IPCP capabilities of novice nurses [65]. The ability to engage in interprofessional collaborative practice is a dynamic development for learners and practitioners, with each competency continuously evolving throughout an individual’s professional career and being honed in the ever-changing practice/learning environment [66]. Therefore, prospective nursing undergraduate teachers, curricula, and newly graduated clinical trainees should implement interprofessional simulation training programs to enhance clinical performance and effective collaboration in the healthcare system, thus providing quality and safe care.

### 4.5. Internal Consistency Reliability

Regarding reliability testing, the reliabilities of the internal consistency of the three core competencies of the IPCPCS were 0.978, 0.976, and 0.970, respectively, with all values exceeding the high-reliability threshold of 0.8. Scholars have noted that Cronbach’s α is the most commonly referenced value, with higher values indicating better reliability, which is generally recommended to maintain values above 0.70 [67]. Thus, the scale used in this study had good internal consistency, demonstrating the reliability of the scale.

### 4.6. Limitations

This study has some limitations. First, this study was a preliminary psychometric validation of a questionnaire. Although the sample size was sufficient for exploratory factor analysis (EFA), the sample size was inadequate for use in confirmatory factor analysis (CFA) testing [68,69]. Future research could conduct CFA to confirm the domains and the overall model fit to determine construct validity. Second, the fact that the overwhelming majority of participants were female could limit the generalizability of the findings when applied to a more gender-diverse population. Third, the self-reported nature of the questionnaire could introduce bias, as participants might provide socially desirable answers. Fourth, the study’s cross-sectional design also prevents causal inferences, and the specific context of Taiwan’s healthcare system may limit the applicability of its findings in different cultural or healthcare settings. Finally, the 38-item questionnaire might affect participants’ willingness to complete the scale and the authenticity of their responses.

## 5. Conclusions

This study has demonstrated the IPCPCS’s validity and reliability in assessing the competencies essential to achieving effective interprofessional collaboration. The three identified core competencies, including collaborative leadership and interprofessional conflict resolution (CLICR), interprofessional communication and team functioning (ICTF), and role clarification and client-centered care (RCCC), explained 77.76% of the total cumulative variance, providing a comprehensive framework for understanding and enhancing interprofessional collaboration among clinical nurses.

Given the scale’s robust psychometric properties, it can serve as a valuable tool in both educational and clinical settings, assisting in efforts to foster a culture of collaborative practice. Future research should conduct criterion-related validity, convergent validity, and discriminant validity studies, and the development of a shorter IPCPC scale for use in nursing is needed. There is also a need for qualitative analysis and long-term tracking to better understand the trends of nurses in IPCP in overall nursing, which may confirm the teaching and learning effectiveness of interprofessional education (IPE). Moreover, we should aim to validate the IPCPCS across different contexts and explore interventions to improve IPCPCS scores, thereby enhancing patient care quality through improved interprofessional collaboration.

In conclusion, the IPCPCS offers a theoretically grounded and empirically validated tool for use in measuring and improving interprofessional collaboration competencies, paving the way for more effective, patient-centered healthcare delivery.

## Figures and Tables

**Figure 1 healthcare-12-00806-f001:**
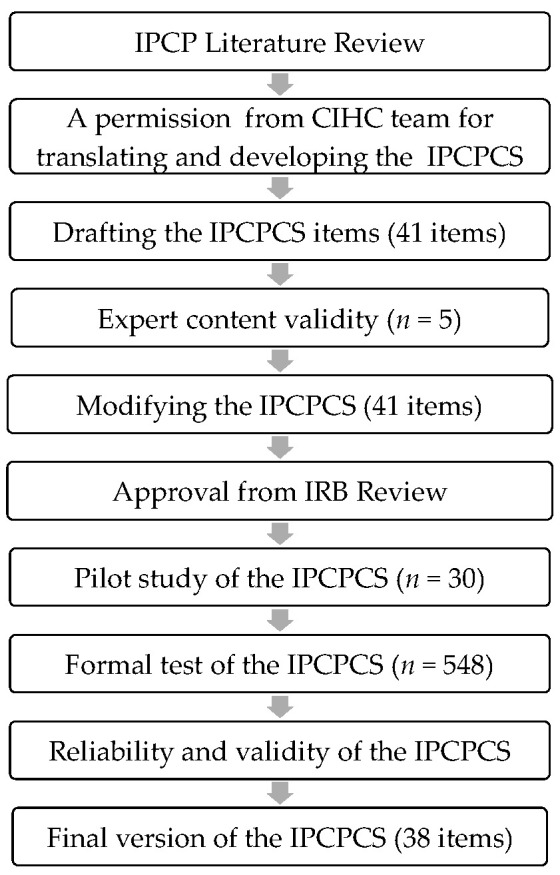
Study procedure.

**Figure 2 healthcare-12-00806-f002:**
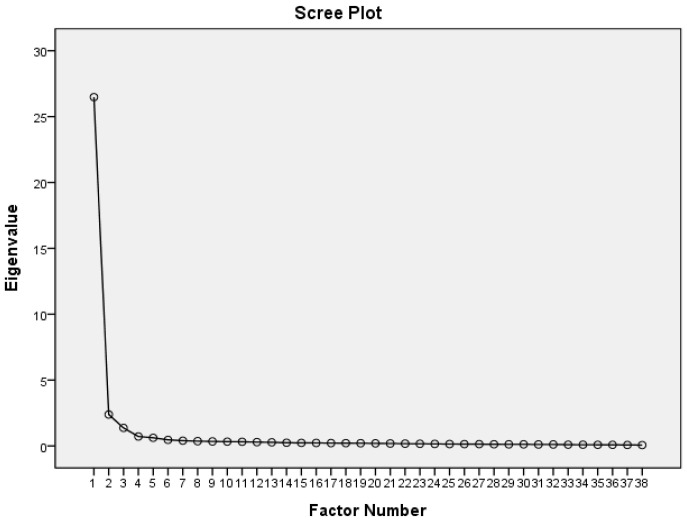
Scree plot of the IPCPCS.

**Table 1 healthcare-12-00806-t001:** Sociodemographic and professional characteristics of the participants (*n* = 548).

Variable	*n* (%)	Range (Median)	Mean (SD)
Gender			
Female	530 (96.7)		
Male	18 (3.3)		
Age (full)		22–60 (33.00)	34.48 (7.87)
20–25	77 (14.1)		
26–30	124 (22.6)		
31–35	117 (21.4)		
36–40	100 (18.2)		
41–45	77 (14.1)		
46–60	53 (9.7)		
Educational level			
Occupational high school/ Junior college	56 (10.2)		
Bachelor’s degree	462 (84.3)		
≥Master’s degree	30 (5.5)		
Hospital district			
A	71 (13.0)		
B	38 (6.9)		
C	19 (3.5)		
D	14 (2.6)		
E	104 (19.0)		
F	189 (34.5)		
G	113 (20.6)		
Years of working experience		0–35.1 (9.50)	11.08 (7.80)
<5 years	133 (24.3)		
5.1–10 years	164 (29.9)		
10.1–15 years	106 (19.3)		
15.1–20 years	62 (11.3)		
≥21 years	83 (15.1)		
Nursing clinical ladder			
≤N1	67 (12.2)		
N2	137 (25.0)		
N3	122 (22.3)		
N4	108 (19.7)		
AHN	52 (9.5)		
HN	50 (9.1)		
NP	12 (2.2)		
Qualified as a clinical instructor?			
No	141 (25.7)		
Yes	407 (74.3)		
Has ever joined interprofessional education (IPE) training?			
No	268 (48.9)		
Yes	280 (51.1)		

Note. N1—clinical work experience of over one year, completion of N1 clinical professional training, passed the review of N1 qualification, and performed patients’/clients’ basic care. N2—clinical work experience of over two years, completion of N2 clinical professional training, passed the review of N2 qualification, and performed critical patients’/clients’ care. N3—clinical work experience of over three years, completion of N3 clinical professional training, passed the review of N3 qualification, performed critical patients’/clients’ holistic care, had teaching and learning ability, and assisted in quality improvement in the working unit; N4, clinical work experience of over four years, completion of N4 clinical professional training, passed the review of the N4 qualification, performed critical patients’/clients’ holistic care, had teaching and learning ability, participated in administration, and performed quality improvement in the working unit [58]; AHN, assistant head nurse; HN, head nurse; NP, nurse practitioner.

**Table 2 healthcare-12-00806-t002:** Total variance explained by the three factors of the IPCPCS (*n* = 548).

Factor	Initial Eigenvalues			Extraction Sums of Squared Loadings		
	Total	Variance (%)	Cumulative Variance (%)	Total	Variance (%)	Cumulative Variance (%)
Factor 1: collaborative leadership and interprofessional conflict resolution (CLICR)	26.48	69.69	69.69	26.26	69.11	69.11
Factor 2: interprofessional communication and team functioning (ICTF)	2.38	6.27	75.96	2.16	5.69	74.79
Factor 3: role clarification and client-centered care (RCCC)	1.37	3.60	79.55	1.13	2.97	77.76

**Table 3 healthcare-12-00806-t003:** Pearson correlation coefficients between the three factors of the IPCPCS (*n* = 548).

Factor	F1	F2	F3
Factor 1: collaborative leadership and interprofessional conflict resolution	1.00		
Factor 2: interprofessional communication and team functioning	0.848 **	1.00	
Factor 3: role clarification and client-centered care	0.814 **	0.883 **	1.00

Note. ** *p* < 0.01 level (2-tailed).

## Data Availability

The data are not publicly available due to ethical restrictions.

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
