# Peer review of "Development and Validation of the Interprofessional Collaboration Practice Competency Scale (IPCPCS) for Clinical Nurses"

_healthcare, 2024, doi:10.3390/healthcare12070806_

Round 1

Reviewer 1 Report

Comments and Suggestions for Authors

1. Please add more keywords (up to 10) ans sort them alphabetically.

2. Add an abbreviation of the scale into a title.

3. Abbreviations were used inattentively, e.g., some of them were introduced twice, i.e., IPCP (lines 41, 47, 51 etc.). In results section: TF1, TF3 etc.

4. Typos, e.g., lines 57.

5. Long paragraphs are unwanted, especially in methodological section. Please reconsider.

6. Lines 188-189: Please indicate clearly the name of research committee.

7. Table 1: age =>46 change into 46-60. Region: what is it? Describe all abbreviations in the table in the notes for this table.

8. Lines 241 and 244 are not informative. Clearly indicate what differences and between what groups are here.

9. Table A1: add mean, SD, skewness and kurtosis, min and max value for all items.

10. Please use CFA in order to support your structure. EFA is not sufficient for this type of work and this high-quality journal.

11. Lines 255-260: Do not repeat in text the information presented in the table. Descriptive statistics for factors can be added into Table A1. Reconsider.

12. The discussion section seems like results section with repeating the results. Please synthesize your findings without repeating.

12. The discussion section is poor. Demographic differences were not described clearly in the results section as well as in the discussion.

13. Please add an appendix or Supplementary Materials with a scale and its scoring keys.

The paper should be very extensively reconsidered. In the current form, it is unclear. Many results and methodological considerations are unspecific.

Author Response

Point 1: 1. Please add more keywords (up to 10) and sort them alphabetically.

Response 1: Keywords have been revised (p.1, Lines 38-39).

Point 2: 2. Add an abbreviation of the scale into a title.

Response 2: An abbreviation has been added ti the title (p.1, Line 3).

Point 3: Abbreviations were used inattentively, e.g., some of them were introduced twice, i.e., IPCP (lines 41, 47, 51 etc.). In the results section: TF1, TF3 etc.

Response 3: Abbreviations of IPCP have been checked and revised in the introduction and results (pp.1-2, Lines 41, 45, 47, 51, 54, 55, 58, 67, 69, & 71).

Point 4: Typos, e.g., lines 57.

Response 4: The introduction has been reviewed and revised again. (p.1, Line 56).

Point 5: Long paragraphs are unwanted, especially in the methodological section. Please reconsider.

Response 5: The methodological section has been reviewed and revised (pp.2-3, Lines 87-144).

Point 6: Lines 188-189: Please indicate clearly the name of the research committee.

Response 6: The name of the research committee has been indicated (p.4, Lines 149-150).

Point 7: Table 1: age =>46 change into 46-60. Region: what is it? Describe all abbreviations in the table in the notes for this table.

Response 7: Categories of age, region, and abbreviations in Table 1 have been revised (p.5-6, Line 181-194) or added to the notes (p.6, Line 182-194). In addition, considering participants’ confidentiality, we used hospital districts A to G for coding from northern to southern Taiwan instead of using real regions (p.5, Line 181).

Point 8: Lines 241 and 244 are not informative. Clearly indicate what differences and between what groups are here.

Response 8: Table A2 has been added to the text (pp.13-14, Line 369)

Point 9: Table A1: add mean, SD, skewness and kurtosis, min and max value for all items.

Response 9: Mean, SD, minimum, and maximum values have been added to Table A1 except for skewness and kurtosis. We never see an EFA table with skewness and kurtosis because they are a part of assumption checking rather than EFA results. Furthermore, the principle of making a table is concise with the necessary information of this study (pp.11-13, Lines 363-368).

Point 10: Please use CFA in order to support your structure. EFA is not sufficient for this type of work and this high-quality journal.

Response 10: According to the literature, EFA can be used to explore patterns underlying a data set. CFA is used to confirm a previously stated theoretical model. Essentially, when using CFA, the researcher is testing whether the data collected supports a hypothesized model. CFA is suitable when the theoretical constructs are well understood and articulated and the validity evidence on the internal structure of the scale (the relationship between the items) has already been obtained in similar contexts [1]. (p.9, Lines 318-320)

As this study is in the early stages of instrument development. By using EFA, it is possible to identify items that empirically do not belong to the expected construct and should be removed from the survey. Furthermore, due to the limitation of sample size [1, 2], the initial sample cannot be randomly divided into two independent groups to complete CFA. Given these two different tools, EFA and CFA must be performed on different data sets; otherwise, overfitting may occur [2]`; therefore, this study was unable to conduct CFA at this time. This is also a limitation of this study and serves as a future research direction.

1.Knekta E, Runyon C, Eddy S. One Size Doesn’t Fit All: Using Factor Analysis to Gather Validity Evidence When Using Surveys in Your Research. CBE—Life Sciences Education. 2019,18(1), rm1.

2.Gunawan J, Marzilli C, Aungsuroch Y. Establishing appropriate sample size for developing and validating a questionnaire in nursing research. Belitung Nursing Journal. 2021, 7(5), 356-360.

Point 11: Lines 255-260: Do not repeat in text the information presented in the table. Descriptive statistics for factors can be added into Table A1. Reconsider.

Response 11: Table 4 and its results and discussion have been deleted (p.7, Line 226).

Point 12: The discussion section seems like results section with repeating the results. Please synthesize your findings without repeating.

Response 12: The discussion has been reviewed and revised again (pp.7-9 , Lines 224-312).

Point 13: The discussion section is poor. Demographic differences were not described clearly in the results section as well as in the discussion.

Response 13: The discussion has been reviewed and revised again (pp.7-9, Lines 227-316).

Point 14: Please add an appendix or Supplementary Materials with a scale and its scoring keys.

The paper should be very extensively reconsidered. In the current form, it is unclear. Many results and methodological considerations are unspecific.

Response 14: Supplementary Materials has been revised (Supplementary, Table S1, S2).

Reviewer 2 Report

Comments and Suggestions for Authors

Dear Authors,

thank you for the opportunity to review your manuscript. Your manuscript needs to be improved. I will list some useful initial suggestions to improve your work here:

1. In the abstract, when you say "There is no Interprofessional Collaboration Scale of Holistic Practice Competency 23 (IPCSHPC) for nursing staff in the world." (line 23), It would be better to say that there is no scale aimed at measuring the outcome you want to measure, more generally; however, it is not true (look at point 2);

2. The rational is not adequately addressed. In the literature, there are attempts, albeit recent, to develop a scale for physicians, nurses and public health professionals, like the one you propose. I attach the bibliographical reference of the authors: Prasitanarapun, R., Kitreerawutiwong, N. The development of an instrument to measure interprofessional collaboration competency for primary care teams in the district health system of health region 2, Thailand. BMC Prim. Care 24, 55 (2023). https://doi.org/10.1186/s12875-023-02013-9. In this sense, your work should bring something original to the themes already dealt with in the literature;

3. In the introduction, the logic of the statements is very fragmented, often not consequential and the information is complex to infer. Too often references are made to other concepts and the reader fails to follow the speech (for example, from line 57 to 72, when you refer to leadership, but the main topic is interprofessional collaborative practice);

4. The statements contained in the text from line 110 to line 114 are not real. Please modify your affirmations by verifying the general literature.

5. The concept of variable and instrument are interchanged, offering little understanding to the reader (line 136).

6. Some limitations of the study, properly identified by you, are difficult to accept, as the generalizability of the results on the basis of the sampling carried out and the instrument administered.

7. I have noticed that in this manuscript very similar forms and phrases are often adopted compared to the general literature on the subject, but the authors are not cited, for example from line 51 to line 56 (Green BN, Johnson CD. Interprofessional collaboration in research, education, and clinical practice: working together for a better future. J Chiropr Educ. 2015 Mar;29(1):1-10. doi: 10.7899/JCE-14-36. Epub 2015 Jan 16. PMID: 25594446; PMCID: PMC4360764). To avoid plagiarism, the text must be fully controlled in its current form.

Comments on the Quality of English Language

It would be advisable to revise the form of English used, to make text more fluent and less redundant.

Author Response

Point 1: In the abstract, when you say "There is no Interprofessional Collaboration Scale of Holistic Practice Competency 23 (IPCSHPC) for nursing staff in the world." (line 23), It would be better to say that there is no scale aimed at measuring the outcome you want to measure, more generally; however, it is not true (look at point 2);

Response 1:. The abstract has been reviewed and revised (p.1, Line 24).

Point 2: The rational is not adequately addressed. In the literature, there are attempts, albeit recent, to develop a scale for physicians, nurses and public health professionals, like the one you propose. I attach the bibliographical reference of the authors: Prasitanarapun, R., Kitreerawutiwong, N. The development of an instrument to measure interprofessional collaboration competency for primary care teams in the district health system of health region 2, Thailand. BMC Prim. Care 24, 55 (2023). https://doi.org/10.1186/s12875-023-02013-9. In this sense, your work should bring something original to the themes already dealt with in the literature;

Response 2: The Introduction has been reviewed and revised (pp.1-2, Lines 54-86).

Point 3: In the introduction, the logic of the statements is very fragmented, often not consequential and the information is complex to infer. Too often references are made to other concepts and the reader fails to follow the speech (for example, from line 57 to 72, when you refer to leadership, but the main topic is interprofessional collaborative practice);

Response 3: The Introduction has been reviewed and revised (pp.1-2, Lines 41-86).

Point 4: The statements contained in the text from line 110 to line 114 are not real. Please modify your affirmations by verifying the general literature.

Response 4: The Introduction has been reviewed and revised (p.2, Lines 76-81 ).

Point 5: The concept of variable and instrument are interchanged, offering little understanding to the reader (line 136).

Response 5: Instruments have been reviewed and revised (p.3, Lines 100; Supplemental Table S2, pp.5-6).

Point 6: Some limitations of the study, properly identified by you, are difficult to accept, as the generalizability of the results on the basis of the sampling carried out and the instrument administered.

Response 6: Limitations have been reviewed and revised (p.9, Lines 317-328).

Point 7: I have noticed that in this manuscript very similar forms and phrases are often adopted compared to the general literature on the subject, but the authors are not cited, for example from line 51 to line 56 (Green BN, Johnson CD. Interprofessional collaboration in research, education, and clinical practice: working together for a better future. J Chiropr Educ. 2015 Mar;29(1):1-10. doi: 10.7899/JCE-14-36. Epub 2015 Jan 16. PMID: 25594446; PMCID: PMC4360764). To avoid plagiarism, the text must be fully controlled in its current form.

Response 7: The manuscript has been reviewed and revised (p.1, Lines 41-44 ).

Round 2

Reviewer 1 Report

Comments and Suggestions for Authors

There are many typos in the paper, e.g., "Orchard et al. (2018) developed the Interprofessional Team Collaboration Scale (AITCS-II), whichOrchard et al. includes three subscales: partnership, cooperation, and coordination [18]."

Please reread paper very carefully.

There are many inconsistencies in using zeros befro full stops in numbers. Please reread and correct.

Author Response

Point 1: There are many typos in the paper, e.g., "Orchard et al. (2018) developed the Interprofessional Team Collaboration Scale (AITCS-II), which Orchard et al. includes three subscales: partnership, cooperation, and coordination [18]."

Please reread paper very carefully.

There are many inconsistencies in using zeros before full stops in numbers. Please reread and correct.

Response 1: We have reviewed and revised the manuscript and supplement again.

Reviewer 2 Report

Comments and Suggestions for Authors

Dear Authors,

unfortunately, the rational is not yet adequately supported by literature. When you say: "Despite the importance of IPCP in healthcare, Taiwan's medical environment currently lacks a structured and standardized tool to assess the IPCP competencies required for clinical nurses." (lines 54-56) and "There is no Interprofessional Collaboration Scale of Holistic Practice Competency (IP-CSHPC) for clinical nurses in Taiwan."(lines 23-24), at the same time, there are many scales and instruments that can measure the phenomenon being studied and which could be used in clinical practice and research, also in Taiwan. The information are still very fragmented and not clear, often not consequential and complex to understand. Some limitations of the study are difficult to accept, as the generalizability of the results on the basis of the sampling and the instrument administered. I can still note some forms of plagiarism relating to the text of some authors who have published similar works on the same scientific theme. I suggest to you a complete revision of the manuscript, paying attention to these comments and making the text more linear and comprehensible.

Comments on the Quality of English Language

A major revision of the English language is required to avoid redundancy of concepts.

Author Response

Point 1: unfortunately, the rational is not yet adequately supported by literature. When you say: "Despite the importance of IPCP in healthcare, Taiwan's medical environment currently lacks a structured and standardized tool to assess the IPCP competencies required for clinical nurses." (lines 54-56) and "There is no Interprofessional Collaboration Scale of Holistic Practice Competency (IP-CSHPC) for clinical nurses in Taiwan."(lines 23-24), at the same time, there are many scales and instruments that can measure the phenomenon being studied and which could be used in clinical practice and research, also in Taiwan. The information are still very fragmented and not clear, often not consequential and complex to understand. Some limitations of the study are difficult to accept, as the generalizability of the results on the basis of the sampling and the instrument administered. I can still note some forms of plagiarism relating to the text of some authors who have published similar works on the same scientific theme. I suggest to you a complete revision of the manuscript, paying attention to these comments and making the text more linear and comprehensible.

Response 1: We have reviewed and revised the manuscript and supplement again. In addition, we have used ithenticate software to detect plagiarism issue. The percentage is 5-10%.
